# The Enzyme-Modified Neutral Comet (EMNC) Assay for Complex DNA Damage Detection

**DOI:** 10.3390/mps4010014

**Published:** 2021-02-16

**Authors:** Maria Rita Fabbrizi, Jonathan R. Hughes, Jason L. Parsons

**Affiliations:** 1Cancer Research Centre, Department of Molecular and Clinical Cancer Medicine, University of Liverpool, 200 London Road, Liverpool L3 9TA, UK; Maria.Fabbrizi@liverpool.ac.uk (M.R.F.); jonathan.hughes@liverpool.ac.uk (J.R.H.); 2Clatterbridge Cancer Centre NHS Foundation Trust, Clatterbridge Road, Bebington CH63 4JY, UK

**Keywords:** comet assay, DNA damage, complex DNA damage, DNA repair, ionising radiation, protons

## Abstract

The comet assay is a versatile, simple, and sensitive gel electrophoresis–based method that can be used to measure and accurately quantify DNA damage, particularly single and double DNA strand breaks, in single cells. While generally this is used to measure variation in DNA strand break levels and repair capacity within a population of cells, the technique has more recently been adapted and evolved into more complex analysis and detection of specific DNA lesions, such as oxidized purines and pyrimidines, achieved through the utilization of damage-specific DNA repair enzymes following cell lysis. Here, we detail a version of the enzyme-modified neutral comet (EMNC) assay for the specific detection of complex DNA damage (CDD), defined as two or more DNA damage lesions within 1–2 helical turns of the DNA. CDD induction is specifically relevant to ionizing radiation (IR), particularly of increasing linear energy transfer (LET), and is known to contribute to the cell-killing effects of IR due to the difficult nature of its repair. Consequently, the EMNC assay reveals important details regarding the extent and complexity of DNA damage induced by IR, but also has potential for the study of other genotoxic agents that may induce CDD.

## 1. Introduction

Over the past decade, the single-cell gel electrophoresis, or comet assay, has become one of the standard methods for assessing DNA damage, with applications in genotoxicity testing, human biomonitoring, and molecular epidemiology, as well as fundamental research in DNA damage and repair, mainly due to its simplicity, sensitivity, versatility, speed and cheapness. When first devised, its simple approach consisted of embedding cells in an agarose matrix on a microscope slide and lysing the cells with non-ionic detergent and high-molarity sodium chloride. This causes the removal of membranes, cytoplasm and nucleoplasm, and the disruption of nucleosomes, leaving an intact nuclear matrix or scaffold composed of ribonucleic acid and proteins, with the DNA wrapped around it in its supercoiled form [1]. When the negative DNA supercoiling is subsequently unwound by the relatively neutral pH buffer (pH = 9.5), this causes the loops expanded out from the nucleoid core following gel electrophoresis to form a comet tail, visualized using ethidium bromide staining and fluorescence microscopy. This seems to be simply a halo of relaxed loops pulled to one side by the electrophoretic field [2]. The procedure was then modified with the use of high pH treatment (>pH 13) pre- and during electrophoresis, which is necessary to reveal DNA single-strand breaks (SSBs) [3,4]. Since its establishment, the comet assay has proven to be one of the most versatile methods for studying cellular DNA repair capacity, as it allows to quantitatively measure the actual DNA damage induced, as well as the damage remaining at intervals after treatment, thus allowing a study of the kinetics of cellular repair [5,6]. This consequently can avoid the interference of other cellular processes, such as antioxidant enzyme activity, cell cycle progression and apoptosis, which can play a role depending on the nature of the damaging agent. Since the study of the kinetics of DNA damage repair can be quite laborious and time-consuming using traditional comet assay techniques, we have recently described a variation of this method by treating cells in a suspension of medium with a genotoxin, embedding the cells within an agarose matrix, and then allowing the cells to repair the DNA damage in situ in a humidified chamber [7], which will be described herein.

Measuring just DNA strand breaks with the conventional comet assay gives limited information on the total and type of DNA damage induced, as the direct effect of some damaging agents will be the generation of modified bases (e.g., oxidative DNA damage), but also sites of base loss (apurinic/apyrimidinic or AP sites) that are alkali labile and therefore appear as breaks under alkaline comet assay conditions (pH > 13). AP sites will also be generated as intermediates during base excision repair of DNA base damage [8]. Moreover, it is generally accepted that, by performing the comet assay using neutral pH conditions (pH = 8), only DNA double strand breaks (DSBs) are detected, and there is plentiful published evidence to support this. This, though, is still a topic for debate [9]. Nevertheless, given that the comet assay largely detects DNA strand breaks, to make the assay more specific as well as more sensitive, an extra step of digesting the nucleoids with an enzyme that recognizes a particular kind of damage, thus creating additional strand breaks, has been introduced. Among the enzymes to be employed, endonuclease III is used to detect oxidized pyrimidines [10], while formamidopyrimidine DNA glycosylase (Fpg) can digest the major purine oxidation product 8-oxoguanine, as well as other altered purines [11]. Additionally, T4 endonuclease V recognizes UV-induced cyclobutane pyrimidine dimers [12], and Alk A incises DNA at 3-methyladenines [13], which can be employed in the EMNC assay. However, most studies have incorporated these enzymes individually into the alkaline version of the comet assay to focus on detecting one single type of DNA damage, without considering the possible presence of complex DNA damage (CDD). CDD is defined as multiple DNA damage types, including DNA base damage, abasic sites, and strand breaks (SSBs and DSBs), generated within close proximity (1–2 helical turns) of the DNA. CDD is largely a signature of ionising radiation (IR) due to the deposition of energy along radiation tracks, and the frequency and complexity of the damage increases with increasing linear energy transfer (LET). Therefore, low-LET x-ray or γ-rays produce lower levels of CDD than higher LET radiation, including protons and carbon ions. Nevertheless, CDD is well-established as a major challenge to the DNA repair machinery, and contributes significantly to the cell-killing effects of IR, although the precise cellular mechanisms and proteins required for its repair are unclear [14,15,16]. Therefore, CDD is an important determinant and factor that requires quantitative analysis in response to IR, but potentially other genotoxic agents as well.

Here, we present a modified version of the neutral comet assay, which includes the use of three different human DNA repair enzymes, AP endonuclease 1 (APE1), 8-oxoguanine DNA glycosylase 1 (OGG1), and endonuclease III homologue 1 (NTH1). This method has been optimized on mammalian cell lines, but it has the potential for adaptation and use with other mammalian cell types, particularly those frequently employed in conventional comet assays (e.g., in peripheral blood mononuclear cells). The use of cells from other animal and non-animal species, including plants that require specific cell lysis conditions to isolate the nuclei, within enzyme modified comet assays has also been reported (see more comprehensive reviews on this topic [17,18]). However, we would recommend that the assay is thoroughly validated and optimized using the specific cell type of interest. Nevertheless, we describe here a revised and up-to-date version of a technique previously described using *E.coli* enzymes [19], and which incorporates additional drying steps and incubation of cells in situ to repair DNA damage [7], as mentioned above. This assay will specifically measure CDD that is DSB-associated, since the assay cannot be performed under alkaline conditions, as only residual DNA base damage above those of SSBs and alkali labile sites generated in the absence of the enzymes will be measured. APE1 catalyzes the cleavage of the phosphodiester backbone at AP sites via hydrolysis, leaving a one-nucleotide gap with 3’-hydroxyl and 5’-deoxyribose phosphate (dRP) termini, OGG1 is involved in the excision repair of predominantly 8-oxoguanine (8-oxoG) and NTH1 excises oxidized pyrimidines (e.g., thymine glycol and 5-hydroxycytosine) from DNA. The use of these three enzymes, particularly in combination, allows the conversion of DNA base damage–associated CDD into additional DNA DSBs, and ultimately, an increase in the visible tail intensity following electrophoresis. 

## 2. Experimental Design

### 2.1. Materials

Library efficient DH5a competent bacterial cells (Fisher Scientific, Loughborough, UK; Cat No.: 11573117)Rosetta2 (DE3) pLysS bacterial cells (Merck-Millipore, Watford, UK; Cat No.: 71403)Bacterial expression plasmid for His-tagged APE1 (e.g., Addgene, Teddington, UK; Cat No.: 70757; or available on request from authors)Bacterial expression plasmid for His-tagged OGG1 (available on request from authors)Bacterial expression plasmid for His-tagged NTH1 (available on request from authors)Agar granules (Fisher Scientific, Loughborough, UK; Cat No.: 10572775)IPTG (Sigma-Aldrich, Dorset, UK; Cat No.: I6758-5G)Glucose (Sigma-Aldrich, Dorset, UK; Cat. No.: 16325-1KG)LB Broth, Miller granulated (Fisher Scientific, Loughborough, UK; Cat No.: 11345992)Lysozyme from chicken egg white (Sigma-Aldrich, Dorset, UK; Cat No.: L4919-1G)Novex™ WedgeWell™ 10% SDS-PAGE gel (Fisher Scientific, Loughborough, UK; Cat No.: 15496794)Leupeptin (Fisher Scientific, Loughborough, UK; Cat No.: 11884101)Chemostatin (Sigma-Aldrich, Dorset, UK; Cat No.: 230790-10MG)Pepstatin (Fisher Scientific, Loughborough, UK; Cat No.: 10036263)Aprotinin (Fisher Scientific, Loughborough, UK; Cat No.: 11854101)PMSF (Fisher Scientific, Loughborough, UK; Cat No.: 10485015)Imidazole (Sigma-Aldrich, Dorset, UK; Cat. No.: I2399-500G)TGS, 10x (Bio-Rad Laboratories, Watford, UK; Cat No.: 161-0772)TG, 10x (Bio-Rad Laboratories, Watford, UK; Cat No.: 161-0771)Methanol (Fisher Scientific, Loughborough, UK; Cat No.: 10675112)Mercaptoethanol (Sigma-Aldrich, Dorset, UK; Cat. No.: M6250-250ML)Glycerol (Fisher Scientific, Loughborough, UK; Cat No.: 10795711)SDS (Sigma-Aldrich, Dorset, UK; Cat. No.: L5750-500G)Bromophenol blue (Fisher Scientific, Loughborough, UK; Cat No.: 10497573)Anti-HisTag antibody (Merck-Millipore, Watford, UK; Cat No.: 70796-3)Instant Blue Protein Stain (Sigma-Aldrich, Dorset, UK; Cat. No.: ISB1L)Agarose low melting point (Fisher Scientific, Loughborough, UK; Cat. No.: 10583355)Agarose normal melting point (Fisher Scientific, Loughborough, UK; Cat. No.: 10688973)PBS (Ca^2+^ and Mg^2+^ free) (Sigma-Aldrich, Dorset, UK; Cat. No.: D8537-500ML)NaCl (Sigma-Aldrich, Dorset, UK; Cat. No.: 31434-1KG-M)NaOH (Sigma-Aldrich, Dorset, UK; Cat. No.: 30620-1KG-M)EDTA disodium salt solution (Sigma-Aldrich, Dorset, UK; Cat. No.: E5134-500G)Tris base (Sigma-Aldrich, Dorset, UK; Cat. No.: T1503-1KG)Triton X-100 (Fisher Scientific, Loughborough, UK; Cat. No.: 10717503)N-Lauroylsarcosine (Sigma-Aldrich, Dorset, UK; Cat. No.: L5125-500G)Dimethyl sulphoxide (Sigma-Aldrich, Dorset, UK; Cat. No.: D5879-500ML)Boric acid (Sigma-Aldrich, Dorset, UK; Cat. No.: B7660-1KG)KCl (Sigma-Aldrich, Dorset, UK; Cat. No.: P3911-500G)MgCl_2_ (Sigma-Aldrich, Dorset, UK; Cat. No.: M8266-100G)DTT (Fisher Scientific, Loughborough, UK; Cat. No.: 10592945)BSA (Fisher Scientific, Loughborough, UK; Cat. No.: 12827172)KOH (Sigma-Aldrich, Dorset, UK; Cat. No.: P1767-500G)SYBR Gold (Fisher Scientific, Loughborough, UK; Cat. No.: 10358492)Trypsin/EDTA (for cell collection) (Sigma-Aldrich, Dorset, UK; Cat. No.: T4049-100ML)

### 2.2. Equipment

QIAprep Spin Miniprep Kit (Qiagen, Southampton, UK; Cat No.: 27104)Syringe filter, 0.45 µm (Fisher Scientific, Loughborough, UK; Cat. No.: 15216869)Syringe filter, 1.1 µm (Fisher Scientific, Loughborough, UK; Cat. No.: 15372378)Sonicator (Sonics, Newtown, CT, USA; Cat No.: VCX 130)Nanodrop (Fisher Scientific, Loughborough, UK; Cat. No.: 13-400-518)Rotary Shaker (Kuhner, Birsfelden, Switzerland; Cat No.: SMX1700)HisTrap HP affinity chromatography column (GE Healthcare, Little Chalfont, UK; Cat No.: 17-5247-01)AKTA FPLC (GE Healthcare, Little Chalfont, UK; e.g., Cat No.: 18-1900-26)Oak Ridge tubes (Sigma-Aldrich, Dorset, UK; Cat. No.: T1418-10EA)Superloop, 10 mL (Fisher Scientific, Loughborough, UK; Cat. No.: 11330122)Immobilon FL membrane (Merck-Millipore, Watford, UK; Cat No.: IPFL00010)Superfrosted microscope slides (Fisher Scientific, UK; Cat. No.: 10149870)Coverslips 50 mm × 22 mm (Fisher Scientific, Loughborough, UK; Cat. No.: 12362128)Coverslips 22 mm × 22 mm (Fisher Scientific, Loughborough, UK; Cat. No.: 12333128)Coplin jars (Fisher Scientific, Loughborough, UK; Cat. No.: 10284922)Humidified chamber (Sigma-Aldrich, Dorset, UK; Cat. No.: Z670146-1EA)Comet electrophoresis tank (Appleton Woods, Birmingham, UK; Cat. No.: CS1602)Power supply unit (Fisher Scientific, Loughborough, UK; Cat. No.: 12613546)Fluorescent microscope Olympus BX61 (Olympus, Hamburg, Germany)

### 2.3. Software

Komet 6.0 image analysis software (Andor Technology, Belfast, Northern Ireland)

### 2.4. Solutions 

LB broth media: 12.5 g of LB broth granules were added to 500 mL of dH_2_O (2.5% *w*/*v*) and the pH was adjusted to 7.2 with 5 M of NaOH.LB Agar: 7.5 g of agar granules were added to 500 mL of LB broth (1.5% *w*/*v*).3× SDS-PAGE loading dye: 750 μL of 1 M of Tris-HCl (25 mM, pH = 6.8), 750 μL of 100% mercaptoethanol (2.5%), 3 mL of 10% SDS (1%), 3 mL of 100% glycerol (10%), 1.5 mL of 1 mg/mL of bromophenol blue (0.05 mg/mL), and 60 μL of 500 mM of EDTA (1 mM) were added to 940 μL of dH_2_O. For the working solution, 2:1 protein extract was diluted in 3× SDS dye.Enzyme Purification Lysis Buffer: 25 mM of Tris-HCl (12.5 mL of a 1 M solution, pH = 8.0), 500 mM of NaCl (50 mL of a 5 M solution), 5% Glycerol (25 mL of a 100% solution), and 5 mM of Imidazole (0.170 g) were added to 412.5 mL of dH_2_O. The complete solution was prepared prior to use by adding a mixture of protease inhibitors (30 μL of leupeptin, chemostatin, pepstatin, aprotinin (all 1 mg/mL), and 100 μL PMSF (100 mM)).Enzyme Purification Elution Buffer: 25 mM of Tris-HCl (12.5 mL of a 1 M solution, pH = 8.0), 500 mM of NaCl (50 mL of a 5 M solution), 5% Glycerol (25 mL of a 100% solution), and 500 mM of imidazole (17.02 g) were added to 412.5 mL of dH_2_O. The complete solution was prepared prior to use by adding 100 μL of PMSF (100 mM).Enzyme Storage Buffer: 50 mM of Tris-HCl (12.5 μL of a 1 M solution, pH = 8.0), 50 mM of KCl (12.5 μL of a 1 M solution), 1 mM of EDTA (10 μL of a 0.1 M solution), and 10% glycerol (10 μL of a 100% solution) were added to 955 μL of dH_2_O.Comet Lysis Buffer: NaCl (146.1 g, 2.5 M), EDTA disodium salt (37.2 g, 100 mM), Tris base (1.2 g, 10 mM), and 1 % N-lauroylsarcosine (10 g) were added to 800 mL of dH_2_O. The solution was heated to ~45 °C if necessary. The pH was set to 9.5 by the addition of NaOH (8 g) and 5 M of NaOH dropwise, then adjusted to 1 L and stored at 4 °C. The complete solution was prepared prior to use by adding a mixture of 1 ml of DMSO and 1 mL of Triton X-100 to 98 mL of cold lysis buffer.Comet Electrophoresis Buffer: A 5× TBE solution was prepared by adding 54 g of Tris base, 27.5 g of boric acid, and 20 mL of 0.5 M EDTA (pH = 8.0) to ~800 mL of dH_2_O. The pH was adjusted to 8.3, with volume of 1 L made up and stored at room temperature (RT). For working electrophoresis buffer (1× TBE), 300 mL of 5× TBE with 1200 mL of cold dH_2_O were mixed just before use.Enzyme Activity Buffer: A 10× enzyme buffer solution was prepared by adding Tris base (605.7 mg, 50 mM), KCl (745.5 mg, 100 mM), MgCl_2_ (47.6 mg, 5 mM), EDTA (26.2 mg, 1 mM), DTT (15.4 mg, 1 mM), and BSA (10 mg) to 10 mL of dH_2_O. The pH was set to 8 by the addition of 5 M of KOH dropwise; 1 mL aliquots were stored at −20 °C. For the working enzyme buffer (1×), 1 mL of 10× solution with 9 mL of dH_2_O were mixed just before use.Enzyme Solution: concentrations of enzymes to use depended on the level of purity and activity (see Section 3.5). As a guide, we routinely added 0.6 pmol of APE1, 6.0 pmol of NTH1, and 5.2 pmol of OGG1 diluted in 1x enzyme buffer per treatment.Staining Solution: SYBR Gold 1 was diluted in 20,000 in dH_2_O, pH = 8.0.

## 3. Procedure

### 3.1. Purification and Expression of Recombinant Enzymes

#### 3.1.1. Overexpression of Recombinant His-Tagged APE1, OGG1, and NTH1

Thaw Rossetta2 (DE3) pLysS cells on ice and add 20 µL of cells into the appropriate number of 1.5 mL tubes.Add 1 µL of each bacterial expression plasmid (5 ng/µL) expressing either APE1, OGG1, or NTH1 to each of three tubes, as necessary, and mix carefully by flicking the bottom of the tube.Incubate cells plus plasmid on ice for 5 min, heat shock for exactly 30 s at 42 °C and return to the ice for at least 2 min.Add 500 µL of LB media pre-warmed to 37 °C to each of the tubes containing the cells and incubate on a rotary shaker for 1 h at 37 °C.Plate 50 µL of each mix onto separate pre-prepared 10 cm LB agar plates (containing the appropriate antibiotics) using bacterial spreaders. The remainder of the mix can be stored at 4 °C, in case further required.Invert the plates and incubate overnight in a static incubator set at 37 °C.The following day, select a single bacterial colony, add to 5 mL of LB containing the appropriate antibiotics, and incubate at 37 °C with shaking at 225 rpm overnight. We would advise selecting 2–3 different colonies and incubating in separate tubes to ensure at least one efficiently grown culture.Add 300 µL of overnight culture to feed a 30 mL culture (i.e. 1:100) containing the appropriate antibiotics and 60 µL of glucose (20%), and grow at 37 °C with shaking at 225 rpm until an OD600nm of 0.6–0.8 is achieved (~3 h).Add 30 mL of culture to feed a 300 mL culture (i.e., 1:10) containing the appropriate antibiotics and 600 µL of glucose (20 %), and grow at 37 °C with shaking at 225 rpm until an OD600 nm of 0.6–0.8 is achieved (~1.5 h). Any culture from the original 30 mL culture can be used to create a glycerol stock by removing 150 µL and adding to 50 µL of 50% glycerol, which can then be stored at −80 °C.Induce the 300 mL culture with 1 mM of IPTG (330 µL from 1 M stock) and grow for a further 3 h at 30 °C with shaking.Centrifuge the culture at 8000 rpm for 20 min, remove the supernatant and freeze the bacterial cell pellets at −80 °C.


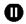
 **PAUSE STEP:** Pellets can be stored indefinitely at −80 °C.

#### 3.1.2. Purification of Recombinant His-Tagged Proteins

Resuspend the bacterial cell pellet thoroughly in 30 mL of complete Enzyme Purification Lysis Buffer.Add lysozyme to 0.1 mg/mL (3 mg) and incubate on ice for 15 min.Lyse the cells by sonication using 3 × 15 s bursts with 30 s intervals on ice.Centrifuge the cell lysate in Oak Ridge tubes at 25,000 rpm for 20 min at 4 °C.Collect the supernatant and filter through 1.1 µm syringe pre-filters, and then through 0.45 µm syringe filters.Prepare 150 mL of the Enzyme Purification Lysis Buffer and 150 mL of the Enzyme Purification Elution Buffer, each containing 0.1 mM PMSF.Wash the 1 mL HisTrap column (usually stored in ethanol) with three volumes of water, followed by three volumes of Enzyme Purification Lysis Buffer containing 0.1 mM PMSF using an FPLC in a cold cabinet/room (4 °C).Add the filtered supernatant to the washed column using a 10 or 50 mL superloop.Wash column with lysis buffer containing 0.1 mM of PMSF until no more protein elutes (~3–5 column volumes).Gradient elute using 20 column volumes (20 mL) of Enzyme Purification Elution Buffer containing 0.1 mM PMSF, collecting 0.5 mL fractions.Remove 5 µL of protein-containing fractions, add 5 µL water and 5 µL 3× SDS PAGE loading dye, and analyze by 10 % SDS-PAGE in 1× TGS Buffer.Transfer proteins to nitrocellulose or PVDF membrane in 1× TG Buffer containing 20% methanol and immunoblot using anti-HisTag antibodies (diluted 1:1000).Use the gel following protein transfer, and stain with Instant blue for > 15 min to identify fractions containing high purity enzyme(s) relative to bacterial contaminantsStore protein fractions at −80 °C until required. Once APE1, OGG1, or NTH1-containing relatively pure (> 90%) protein fractions have been identified, combine these for the individual proteins and buffer exchange or dialyze into the Enzyme Storage Buffer. If proteins are not of sufficient purity, proceed with further purification (e.g., ion exchange chromatography).It is recommended that the activity of the enzymes are checked using oligonucleotide substrates containing the appropriate site-specific DNA damage (e.g., AP site for APE1, 8-oxoguanine for OGG1, and thymine glycol for NTH1; [20]).

### 3.2. Agarose Preparation (10 min)

#### 3.2.1. For Slides Coating

Prepare 1% normal melting point agarose by mixing powdered agarose with distilled water in a glass beaker or bottle.Place bottle in the microwave at low power for short intervals (~30 s), avoiding vigorous boiling of the agarose and ensuring that all of the agarose is dissolved.Agarose solution can be used immediately or stored at RT.

#### 3.2.2. For Cell Embedding

Prepare 1% low melting point agarose by mixing powdered agarose with PBS in a glass beaker or bottle.Place bottle in the microwave at low power for short intervals (~30 s), avoiding vigorous boiling of the agarose and ensuring that all of the agarose is dissolved.Agarose solution can be used immediately (once cooled to 37 °C) or stored at RT.

### 3.3. Slide Coating (10 min)

Prepare slides by adding 800 µL of molten normal melting point agarose to a microscope slide, add a 22 × 50 mm coverslip, and leave agarose to set (~2–5 min) on a flat surface. Remove coverslip, carefully sliding sideward, and air dry slides overnight.

### 3.4. Cell Embedding and Lysis (30 min)

Trypsinise cells and dilute to ~1 × 10^5^ cells/mL. Add 250 µL of cell suspension per well of a 24-well plate on ice to prevent repair and adhesion. Induce DNA damage by chemical or physical stress within plate. Note that chemicals with a long half-life will continue to induce DNA damage in cells in suspension, so these are not recommended to be used using this method. Alternatively, cells can be grown as monolayer and exposed separately to stress according to the experiment design (compound concentrations, time exposure) and then trypsinized. The amount of DNA damaging agent should be determined empirically, but should induce a DNA damage level of <35% tail DNA to ensure this is not too extensive for the cell to repair, if analyzing DNA repair efficiency.Add 500 µL of low melting point agarose (previously melted and maintained at 37 °C) to cells, mix gently, and add 70 µL of this cell suspension to two areas on a normal melting point agarose-coated slide (equivalent to duplicate treatments). Add two 22 × 22 mm coverslips and place on the ice tray for ~2 min for the agarose to set. Two slides per treatment should be prepared (one for the buffer treatment only detecting DSBs, and one for the enzyme treatment detecting both DSBs and CDD).A negative control (no DNA damage treatment) should be prepared. Ideally, the experiment would also include a positive control with a treatment known to induce CDD (e.g., high-LET radiation).If analyzing DNA repair efficiency, place slides in a humidified chamber at 37 °C to allow repair according to experiment time-point (e.g., up to 6 h).After incubation, carefully remove coverslips by sliding sideward and place slides in coplin jars containing fresh cold lysis buffer. Lyse cells for at least 1 h at 4 °C.


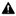
 **CRITICAL STEP:** duration of cell lysis may vary among different cell lines and must be empirically determined.
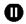
 **PAUSE STEP:** slides can be kept in lysis buffer overnight at 4 °C.

### 3.5. Enzyme Treatment (100 min)

Wash slides three times, for 5 min each, in PBS in coplin jars, at 4 °C (to remove lysis buffer).Lay slides on a flat surface and place 50 µL of enzyme solution per agarose/cell area, or buffer alone for the untreated slides, and cover with a 22 × 22 mm coverslip. Note that the amount of enzyme to use should be predetermined beforehand, ideally by titrating each enzyme (NTH1, OGG1, and APE1) against a positive control (e.g., high-LET radiation) versus a negative control (e.g., low-LET radiation) to ensure that CDD is only revealed largely under the former conditions.Place slides in a humidified chamber and incubate at 37 °C for 1 h to allow enzyme processing.

### 3.6. Electrophoresis (105 min + Overnight Dry)

Carefully remove coverslips by sliding sideward and wash slides three times for 5 min each in coplin jars containing 1× PBS.Transfer slides to an electrophoresis tank (can be placed at 4 °C if ambient temperature is relatively high), organize them into two separate columns, and cover with ~1.2 L of the fresh cold Comet Electrophoresis Buffer.Incubate slides for 30 min.Electrophorese slides at 25 V (1 V/cm, ~20 mA; note that volume of buffer may need to be removed/added to adjust to the correct current) for 25 min.Carefully remove slides from the electrophoresis tank to minimize movement of the slides (and potential loss of agarose/cells) and lay on a flat surface covered with paper roll. Cover agarose/gel areas with cold 1× PBS buffer (~500 µL per slide) for 5 min. Repeat twice.Pour off excess PBS, lay slides flat, and allow to dry overnight.

### 3.7. Rehydration and DNA Staining (75 min + Overnight Dry)

Place dried slides in coplin jars containing dH_2_O (pH = 8.0) for 30 min to rehydrate agarose.Lay slides on a flat surface covered with paper roll and add enough SYBR Gold (diluted 1:20,000) to cover each slide (~500 µL) for DNA staining. Cover the slides to protect from light and incubate for 30 min.Remove excess stain from the slides, lay flat, and allow slides to dry overnight (while protecting from light) prior to analysis or storage in a sealed box.

### 3.8. Analysis

Capture images of stained DNA from the dried slides using a fluorescent microscope equipped with a 10× objective. There is no standard procedure for measuring the intensity of light and correcting the level of DNA migration accordingly, therefore, this should be optimized empirically.Images can be analyzed live or offline using a validated image analysis software and by scoring 50 cells per agarose/cell square across multiple images, which are present in duplicate on each slide. In our experience, we recommend the Komet 6.0 image analysis software, although other commercial and free software is available (e.g., Comet Assay IV, OpenComet, and CometScore).When analyzing images, it is important to recognize the presence of apoptotic cells (also called “hedgehogs”) that have extremely high DNA damage levels (>90% of DNA in the comet tail), and that these should be quantified separately from cells containing levels of DNA damage (0–40 %) that are amenable to cellular repair.

## 4. Expected Results

### 4.1. Principles of the EMNC Assay

Here, we developed and described a protocol for the detection of CDD using the EMNC assay, which incorporates the use of three different recombinant DNA repair enzymes (APE1, OGG1, and NTH1). The neutral version of the classic comet assay provides details regarding the quantitative levels of DNA DSBs and the capacity of cells to repair these, but does not allow an assessment of DNA damage complexity (Figure 1A). Therefore, the addition of an enzyme treatment step post cell lysis will recognize and incise any residual DNA base damage (based on the specificity of the enzyme used), and which, if in close proximity, will create additional DSBs that can be separated by electrophoresis and detected following DNA staining and image analysis (Figure 1B). Usually, the EMNC assay is performed both in the absence and presence of enzymes, therefore, this assay advantageously reveals the relative levels and repair of both DSBs and CDD in a single experiment.

### 4.2. Detection of CDD Following Proton Irradiation

We have validated that the EMNC assay detects CDD using cells irradiated with protons at lower energies, and therefore increasing LET compared to high energy (low-LET) protons. High-LET radiation, including protons at and around the Bragg peak where the radiation is deposited, are well-known to induce increased levels of CDD through the densely ionising track structure, whereas low-LET radiation generates more spacely-separated DNA damage [14]. When the EMNC assay was performed in the absence of an enzyme (APE1, OGG1, and NTH1) treatment, this revealed that the levels and induction of DSBs (shown as % tail DNA) are the same immediately following proton irradiation with cells positioned at the relatively high-LET at the Bragg peak distal end, versus cells positioned at the low-LET entrance (Figure 2A, compare dark blue and dark green bars at time 0; Figure 2B,C). Additionally, the efficient repair of DSBs under both of these conditions is apparent 4 h post-irradiation. In contrast, performing the EMNC assay in the presence of an enzyme (APE1, OGG1, and NTH1) treatment reveals that additional DSBs (corresponding to CDD) are immediately generated only following relatively high-LET protons, and not low-LET protons (Figure 2A, compare light green and light blue bars at time 0; Figure 2B,C). Furthermore, the increased persistence of CDD is shown through the observation that these exist for at least 4 h post-irradiation with relatively high-LET protons, consistent with the theory that CDD represents a challenge to the cellular DNA repair machinery. Note that the levels of DSBs were not significantly increased in unirradiated cells in the presence versus the absence of enzyme treatment (Figure 2A, compare dark blue/green and light blue/green bars at control; Figure 2B,C). More comprehensive analysis has previously been performed [21,22], which has clearly demonstrated that relatively high-LET protons generated at the distal end of the Bragg peak can induce CDD in cells that persists for several hours (>4 h) post-irradiation, and that this contributes significantly to the increased cell-killing effects observed under these conditions.

## 5. Conclusions

CDD is a major factor involved in the cell-killing effects of ionising radiation, as this persists in cells post-treatment and is significantly more difficult to repair than isolated DNA lesions, given that CDD will consist of multiple DNA damage types (e.g., base damage, SSBs and DSBs). However, a quantitative assay to measure and visualize CDD has been a long-standing challenge in the radiobiology field. We demonstrate that the EMNC assay can clearly be used to quantify the levels of CDD (DSB-associated) in cultured cells in the absence and presence of irradiation, and indeed, could be utilized for the assessment of other DNA damaging agents for their ability to induce CDD. However, the EMNC assay can also be employed to establish the capacity of individual cells and cell populations to repair CDD. Therefore, the EMNC is a valuable methodology and resource for quantitatively determining CDD levels following different sources of ionising radiation, particularly high-LET protons and heavy ions, but also to reveal specific details regarding the CDD repair efficiency of different cell models. Additionally, the enzymes and critical DNA repair mechanisms that are responsive to CDD are still debatable [14], and are very much dependent on the radiation source (and LET) itself. To this effect, we have recently demonstrated that CDD induced by relatively high-LET protons triggers a specific cellular DNA damage response mediated by histone H2B ubiquitylation catalyzed by the E3 ubiquitin ligases ring finger protein 20/40 (RNF20/40) and male-specific lethal-2 (MSL2) [21]. We have also utilized an siRNA screen for identifying deubiquitylation enzymes that modulate radiosensitivity following relatively high-LET protons, which revealed a critical role for ubiquitin specific protease (USP6) in this process. Here, we revealed evidence that USP6 is essential for maintaining the stability of the SSB repair protein poly(ADP-ribose) polymerase-1 (PARP-1), and that targeting the PARP-1 protein itself using siRNA, or inhibiting PARP-1-dependent poly(ADP-ribosyl)ation using olaparib, is able to significantly decrease the survival of cells in response to relatively high-LET protons [22]. Consequently, the utilization of the EMNC has allowed us to establish new mechanistic evidence supporting the roles for changes at the histone and chromatin level, as well as in identifying a critical role for the SSB repair pathway in responding to proton-induced CDD, which requires further research and development. 

## 6. Troubleshooting

Causes and solutions to a number of potential problems encountered during execution of the EMNC assay (Table 1).

## Figures and Tables

**Figure 1 mps-04-00014-f001:**
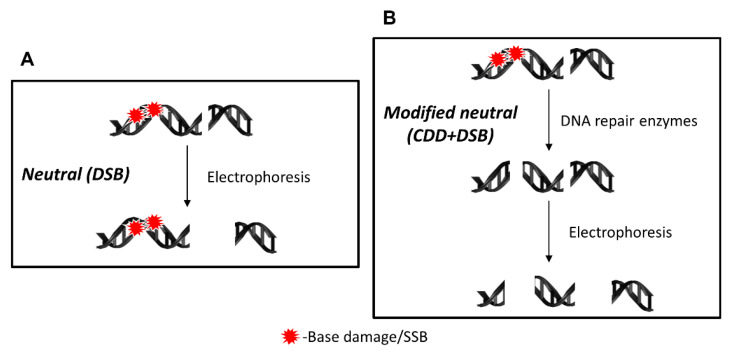
Schematic of the DNA damage detected by different versions of the neutral comet assay. (**A**) Under neutral comet assay conditions, only the analysis and detection of DSBs will be achieved, with no ability to resolve residual DNA base damage and/or SSB in proximity to the DSB. (**B**) In the EMNC assay, following cell lysis, the DNA is treated with recombinant DNA repair enzymes that recognize and incise the DNA at unrepaired DNA base damage sites to create additional DSBs that can then be detected following electrophoresis.

**Figure 2 mps-04-00014-f002:**
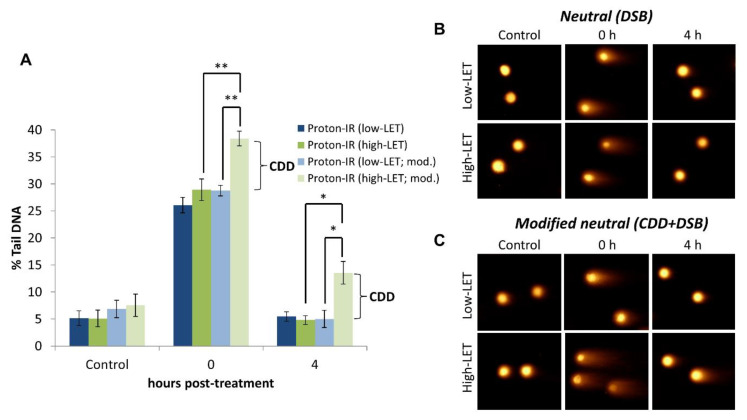
Demonstration of the induction and detection of CDD induced by relatively high-LET protons by the EMNC assay. (**A**) HeLa cells were either un-irradiated (control) or irradiated with 4 Gy protons at relatively high-LET (11 MeV = mean energy) or low-LET (58 MeV), with DNA damage measured immediately, and at 4 h post-irradiation by the EMNC assay. The levels and repair of DSBs are revealed in the absence of enzyme treatment, whereas, following enzyme treatment with APE1, NTH1, and OGG1 (indicated as mod.), this additionally reveals the levels of CDD. Shown is the % tail DNA ± S.D. * *p* < 0.01, ** *p* < 0.0005, as analyzed by a one sample *t*-test. Respective images of cells acquired in (**B**) absence and (**C**) following enzyme treatment pre- and post-irradiation are shown.

**Table 1 mps-04-00014-t001:** Cause and solutions.

Problem	Possible Causes	Remedies
Agarose does not remain attached to the glass slide.	Slides are not adequately pre-coated. Comet Lysis Buffer is not cold enough. Comet Electrophoresis Buffer is not cold enough. Slides are not handled gently before/after lysis and electrophoresis.	Ensure the agarose coating is even and without bubbles Ensure Comet Lysis/Electrophoresis Buffer is cold and kept at 4 °C at all times Electrophoresis step can be performed at 4 °C Always move slides carefully
Cells do not show comets	Cells were not adequately lysed Electrophoresis not effective	Increase the time the cells are left in Comet Lysis Buffer Check electrical connections and voltage/current settings
Cells do not show more extensive comets after enzyme treatment	DNA damaging agent is not inducing CDD The level or activity of the enzymes is insufficient	A positive control (e.g., high-LET radiation) should be used alongside treatment of interest Use more enzyme(s) at the enzyme treatment step Check activity of the enzyme(s) independently
Cells in the untreated (negative) control have large comet tails	Unwanted damage to cells has occurred during sample preparation.	Handle the cells gently and use healthy growing cells Keep the cells on ice between exposure to stress and agarose embedding Reduce trypsin concentration or time of exposure
Cells in the treated samples (particularly positive controls) have very small comet tails	DNA has not migrated sufficiently	Check electrical connections and voltage/current settings Increase electrophoresis time, if necessary
Comets are unevenly distributed across the slides	Comet Electrophoresis Buffer is uneven inside the electrophoresis tank	Use spirit level to ensure the electrophoresis tank is level and all the slides are evenly covered
No, or poorly visible, comets are observed	Insufficient amount of staining	Increase the SYBR Gold concentration, or the time for staining Ensure slides and agarose have been washed thoroughly following electrophoresis
Quality of the comet images is poor	Microscope lens is not clean or focused Poor quality of camera Errors in sample preparation	Keep the microscope lens clean and properly focused Ensure every step in the sample preparation and protocol is properly executed

## Data Availability

The data presented in this study are available on request from the corresponding author.

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
