# Peer review of "The Enzyme-Modified Neutral Comet (EMNC) Assay for Complex DNA Damage Detection"

_mps, 2021, doi:10.3390/mps4010014_

Round 1

Reviewer 1 Report

Fabbrizi et al. presented a protocol that reveals the complex DNA damage. The authors incubated cells with three DNA endonucleases (APE1, OGG1, and NTH1) before unwinding DNA in the neutral buffer. This method enables the detection of additional DNA lesions and would be a useful tool to study DNA damage and repair. Overall, the manuscript is clearly written and topically relevant. I feel that the manuscript is a strong candidate for publication in Methods and Protocols. However, I have a number of minor comments that should be addressed prior to publication.

Comments:

  1. Line 148 and 149, the unit for pore size should be μm not μM.
  2. Line 207 and 328, what is the dilution for SYBR Gold? 1:10,000 or 1:20,000?
  3. In Section 3.6 Electrophoresis: It is not clear to me whether the procedures in this section should be performed at 4°C or not.
  4. In Figure 2B and 2C, it would be helpful to label either neutral comet assay or modified neutral comet assay next to the images.

Reviewer 2 Report

Despite its apparent simplicity, the DNA Comet Assay has many complexities. The protocol developed by the authors is described in sufficient detail and may well be of interest to a certain circle of readers after clarifying some of the details.

  1. The approach proposed by the authors clearly has restrictions on the origin of the material. Does this protocol apply to plant tissue? Or does it only work on human cells? What is the origin of material ? I believe that the application area should be clearly spelled out in the protocol, now this point is absent.
  2. Page 3 Lane 106 Typo
  3. What software except "Komet 6.0" can be used in this protocol?
  4. The protocol does not contain any calibration curves, description of the required positive and negative controls and their meaning under the conditions described. Figure 2 contains only 2 experimental measurements with a dose difference of almost 5 times (11 MeV and 58 MeV), which are not enough for plotting any curves. It is not clear how these doses are chosen and what happens in the range between 11 MeV and 58 MeV.
  5. Paragraph with Troubleshooting is highly desirable in similar manuscripts

Round 2

Reviewer 2 Report

I think that this manuscript could be accepted now. But I still strongly recommend to authors to think well about the applicability of their methodology for non-animal cells, I have a doubts about this
